# Fatty Acids Profile and Antioxidant Properties of Raw Fermented Sausages with the Addition of Tomato Pomace

**DOI:** 10.3390/biom12111695

**Published:** 2022-11-16

**Authors:** Patrycja Skwarek, Małgorzata Karwowska

**Affiliations:** Department of Animal Food Technology, Sub-Department of Meat Technology and Food Quality, University of Life Sciences in Lublin, Skromna 8, 20-704 Lublin, Poland

**Keywords:** tomato pomace, raw fermented sausage, antioxidant properties, fatty acid profile

## Abstract

The aim of the study was to evaluate the effect of tomato pomace (TP) on physicochemical parameters and fatty acid profile as well as antioxidant properties of dry fermented sausages with a reduced content of nitrites. Four different sausage formulations were prepared: control sample, and samples with 0.5%, 1% and 1.5% addition of freeze-dried TP. The sausages were analyzed for: chemical composition, pH and water activity, fatty acid profile, color parameters, biogenic content, and number of lactic acid bacteria and *Enterobacteriacea*. The antioxidant properties were also assessed depending on the amount of TP used. The products were characterized by similar water activity and pH in the range of 0.877–0.895 and 4.55–4.81, respectively. The effect of the addition of freeze-dried TP on an increase in antioxidant activity along with an increase in the concentration of the additive was observed. This phenomenon was most likely due to the strong antioxidant properties of tomato as well as the high content of lycopene. The antimicrobial properties of TP in raw fermented sausages were also noted as the product with the highest concentration of pomace had the lowest number of *Enterobacteriaceae*. In addition, sausages with reduced levels of nitrites to which TP was added were characterized by a higher redness, which will probably have a positive impact on the assessment consumers make of them. The most promising results were obtained for the dry fermented sausage with 1.5% addition of TP.

## 1. Introduction

The level of consumption of meat products on a world scale is constantly increasing. World meat consumption has quadrupled since 1961 per capita [1]. This is mainly due to the modern lifestyle, but also to the fact that most of the other available products are poor in valuable nutrients such as heme iron, protein, and natural antioxidants [2]. The meat sector is therefore considered to be one of the most important in the world [3]. Meat is an essential part of the human diet, mainly due to its nutritional values and organoleptic features. Among the many types of meat products, dry-fermented sausages are of particular importance for consumers, as they are considered to be a healthy and safe food [4]. Fermented meat is included in the human diet as an important component due to its nutritional value, shelf-life and special sensory features [5]. The characteristic aroma as well as the taste that make them unique are considered the most important attributes that strongly influence the acceptance and preferences of modern consumers [6,7]. The popularity of these meat products is related to the growing demand for products with extended shelf life, and fermentation is one of the favored techniques traditionally used for this purpose [8,9]. The shelf life is extended thanks to the use of lactic acid bacteria (LAB) in this technique. These bacteria are responsible for the acidification, desirable taste, color and texture of the products, and for preventing the growth of pathogenic microorganisms. As a result, the stability, safety and shelf life of sausages is improved [7,10,11,12]. The use of fermentation to preserve the meat also avoids the loss of valuable compounds [13,14,15]. As oxidative processes and the growth of pathogenic microorganisms shorten the shelf life of products [16,17], the production of meat products with high storage stability requires the use of antioxidant and antimicrobial additives [18]. In this context, nitrogen compounds are widely used additives in meat processing, primarily as an inhibitor of the growth of certain pathogens; additionally, they give the products a characteristic red-pink color and their unique taste [19]. However, despite the benefits resulting from their use, they are considered unhealthy due to the potential formation of carcinogenic N-nitroso compounds, especially N-nitrosamines [20]. In recent years, thereofre, there has been a growing interest in the elimination or reduction of these synthetic additives, and focusing on production that is towards the “clean label” trend [21]. The increase in people’s awareness of the relationship between diet and health, along with new processing technologies, have both heightened interest in the development of new, healthy food products. Among these, much attention has been paid to the development of functional food products enriched with natural functional bioactive compounds. It has been shown that many bioactive compounds, such as carotenoids, extracted from natural sources, exhibit antimicrobial, antibacterial, antifungal, antiviral, anti-inflammatory or antioxidant properties [22]. Food industry by-products are a good source of natural antioxidants and antimicrobials, and are an alternative to the conventional antioxidants currently used [23].

The tomato is one of the most famous vegetables in the world. It is an excellent source of nutrients such as lycopene, proteins, organic acids and vitamins [24]. Tomatoes are used in the food industry to create a wide variety of products, however it is the peel and seeds that are among the underused parts of tomatoes that contribute to food waste [25]. The processing of tomatoes produces tomato pomace, of which 60% are seeds and 40% are skins [26]. The studies conducted so far indicate the valuable nutritional value of tomato pomace. It has been shown that due to the presence of antioxidant compounds, especially flavonoids, they prevent cancer and cardiovascular diseases [27,28]. There is increasing evidence to support the health benefits of carotenoid consumption [29]. Therefore, obtaining them from tomato skin can be an excellent solution to benefit from them and reduce losses during processing. Direct addition of these by-products to meat products may therefore be a cheaper solution than isolating lycopene from them [30]. Moreover, due to thermal stability and antioxidant properties, tomato seed extracts can be used for food preservation [31,32]. So far, research has been conducted on the use of tomato by-products as an additive to meat products, such as frankfurters, beef patties [33,34] or dry fermented sausages enriched with lycopene [31]. Eyiler and Oztan [35] reported that dried tomato powder reduces the level of nitrite residues in frankfurters and also acts as a natural coloring agent. Wang et al. [36] showed, however, that tomato skin powder proved to be an excellent substitute for fat in sausages. The results showed that the inclusion of tomato skin powder reduced the total amount of animal fat by 0.5–3.0% (*w*/*w*) as well as improved the fatty acid profile and reduced lipid oxidation in the final product. There is also a lot of research in the world literature on the potential of plant by-products as a source of natural antioxidants, which are increasingly used as food additives and functional food ingredients. Therefore, the isolation of these bioactive compounds from by-products such as tomato pomace, which are produced in the agri-food chain, may be one of the promising ways to meet the growing demand for natural antioxidants [37,38]. In this context, the use of tomato pomace in meat products in order to reduce nitrite/nitrate as well as improve the color of the product and increase its antioxidant activity an is innovative and interesting solution that needs to be explored. However, further research is first required to fully understand the role of tomato pomace on sausage stability.

The aim of this study was therefore to evaluate the effect of tomato pomace on physicochemical parameters, fatty acid profile and antioxidant properties of dry fermented sausages with reduced nitrite content. The antioxidant properties of tomato pomace were investigated. The sausages were analyzed after the end of production, including: chemical composition, pH and water activity, fatty acid profile, color parameters (CIE L*a*b*), antioxidant properties against ABTS + (2,2′-Azino-bis-(3-ethylbenzothiazoline-6-sulfonic acid) and DPPH (2,2-Diphenyl-1-picrylhydrazyl) radicals. Microbiological analyses and tests for the presence of biogenic amines were also performed.

## 2. Materials and Methods

### 2.1. Tomato Pomace (TP) Preparation and Analysis

The material for the research was one variety of tomatoes (*Solanum lycopersicum* L.) purchased at a local supermarket. The tomatoes were washed and then pressed in a kitchen press to obtain tomato pomace, which was used as a tomato by-product for further analysis. The pomace of this vegetable was freeze-dried using a freeze dryer (Labconco Free-Zone, USA) at the temperature −50 °C. The dried material was ground in a laboratory mill to obtain a test powder. The dried products were placed in airtight containers and stored at −60 °C. Thus prepared, freeze-dried TP (called TP) was analyzed in accordance with the following methods.

#### 2.1.1. Antioxidant Activity of TP

The antioxidant activity of TP was evaluated in methanolic extracts which were obtained by mixing 5 g of the homogenized TP samples with 100 mL of methanol under constant stirring using an incubator shaker at 150 rpm and 25 °C for 3 h. After that, the extracts were filtered using Whatman No. 1 filter paper (Whatman, Fisher Scientific, Schwerte, Germany). 

##### DPPH Radical Scavenging Activity

The anti-radical ability of TP extracts was evaluated according to Brand-Williams et al. [39] and Vinha et al. [40] with minor modifications. The DPPH radical cation was produced by keeping the DPPH stock solution in the dark at room temperature for 30 min to allow the completion of radical generation. This solution was then diluted with methanol so that its absorbance was adjusted to 0.90 ± 0.02 at 517 nm. TP extracts (300 microliters) were mixed with 2.7 mL of methanolic DPPH. The mixture was shaken and absorbance reading was performed at 517 nm. The DPPH capacity was calculated from a standard curve of Trolox equivalent and expressed as mg per mg. 

##### ABTS^+^ Radical Scavenging Activity

The potential of extracts for radical scavenging was tested using the methanolic solution of ABTS according to Re et al. [41], Tarko et al. [42] and Gaafar et al. [43] with minor modifications. ABTS^+^ (7 mM) was dissolved in methanol and the final solution was diluted in methanol to an absorbance of 0.7 ± 0.02 at 734 nm. ABTS was dissolved in distilled water to a 7 mM concentration. The ABTS radical cation was produced by dosing the ABTS stock solution with 2.45 mM potassium persulfate (final concentration) in the dark at room temperature for 12–16 h to allow the completion of radical generation. This solution was then diluted with methanol so that its absorbance was adjusted to 0.70 ± 0.02 at 734 nm. TP extracts (500 µL) were mixed with 1 mL of methanolic ABTS+. The mixture was shaken and absorbance readings were performed at 734 nm. The ABTS + capacity were calculated from a standard curve of Trolox equivalent and expressed as mg per mg.

#### 2.1.2. Total Phenolics Content (TPC)

The amount of total phenolic compounds was determined according to the procedure described by Vinha et al. [40], Cicco et al. [44] and Azabou et al. [45], with minor modifications. TP samples (5 g each) were subjected to extraction with 100 mL of methanol under constant stirring using an incubator shaker at 150 rpm and 25 °C for 3 h. After that, the extracts were filtered using Whatman No. 1 filter paper (Whatman, Fisher Scientific, Schwerte, Germany). Next, the TP extract (20 μL) was mixed with 100 μL of Folin–Ciocalteu’s reagent. The mixture was shaken and then incubated for 2 min at 25 °C. A volume of 800 μL of sodium carbonate solution (5%, *w*/*v*) was added and the mixture was shaken for 1 min, incubated in the dark for 20 min at 40 °C and then immediately cooled. The absorbance was measured at 760 nm. Gallic acid was used as standard for the analytical curve. TPC was expressed as mg GA equivalents (E)/g extract.

### 2.2. Dry Fermented Sausage Preparation and Analysis

The experimental meat products were manufactured using ham muscles and backfat from Polish large white purebred fatteners obtained from a local slaughterhouse at 48 h postmortem. The study was performed in the Department of Meat Technology and Food Quality (University of Life Sciences in Lublin, Poland) in semi-technical conditions. Pork meat and backfat were used in the proportion of 85:15. Four different sample groups of dry fermented sausages were produced with reduced sodium nitrite addition (50 mg kg^−1^) in relation to the permitted amount in accordance with the Commission Regulation (EU) No. 1129/2011 [46]. The meat was minced through a 0.01 m grinding plate using a commercial grinder (KU2-3EK, Mesko-AGD Skarzysko-Kamienna, Poland). To each formulation, 0.6% of glucose and 2.8% of curing mixture (sea salt + sodium nitrite) were added. Ground tomato pomace (seeds and skins), which had previously been subjected to the freeze-drying process was used in the levels of 0.5, 1.0 and 1.5%. The dried TP was ground just before using a knife mill (Bosch TSM6A017C) for particles less than 0.3 mm in diameter. Four different formulations of the sausages were prepared: SK—control sample; STP 0.5%—sample with 0.5% addition of TP; STP 1%—sample with 1% addition of TP; STP 1.5%—sample with 1.5% addition of TP. All ingredients were mixed using the universal machine type KU2-3EK (Mesko-AGD, Skarzysko-Kamienna, Poland) with an attached R4 type mixer (100 rpm, 3 min). In the next step stuffing were filled into fibrous casings (ø 65 mm, Viskase Corporation, Chicago, IL, USA). Sausages of about 500 g were prepared. Sausages were weighed and hung in a temperature- and humidity-controlled chamber (ITALFROST-DE RIGO-GS, Pszczyna, Poland) until 30 ± 3% weight loss was achieved (17 days). Production conditions consisted of: Stage 1—T 20–22°C, RH 55–65%, 3 days; Stage 2—T 14–16 °C, RH 68–75%, 3 days; and Stage 3—T 13°C, RH 76%, 11 days. Cross-sectional appearance of fermented sausage at the end of production is presented in Figure 1. 

#### 2.2.1. Chemical Composition

The chemical composition of the dry fermented sausages (collagen, moisture, protein, and fat contents) was determined using a Food Scan Lab 78,810 (Foss Tecator Co., Ltd., Molecules 2022, 27, 652 13 of 16 Hillerod, Denmark). Approximately 200 g of each homogenized sample was distributed in the instrument’s round sample dish and loaded into the instrument’s sample chamber.

#### 2.2.2. The Physicochemical Parameters (pH, and Water Activity)

The pH of sausage homogenates was measured with a digital temperature-compensated pH meter (CPC-501, Elmetron, Zabrze, Poland) with a pH electrode (ERH-111, Hydromet, Gliwice, Poland) calibrated with buffer solutions (pH 4.0, 7.0, 9.0). The water activity (a_w_) was measured using a water activity analyzer (Novasina AG, Lachen, Switzerland), which gives temperature-controlled measurements. The analyzer had been calibrated with Novasina SAL-T humidity standards (33%, 75%, 84%, and 90% relative humidity).

#### 2.2.3. Fatty Acid Profile Measurements

The fatty acid profile was determined by gas chromatography after conversion of the fats to fatty acid methyl esters (FAME) [47]. The method of Folch et al. [48] was used for the extraction of lipids from samples. A gas chromatographic analysis was performed using a chromatograph (Varian 450-GC, Walnut Creek, CA, USA) equipped with a capillary column (Select Biodiesel for FAME, Varian, Palo Alto, CA, USA, 30 m × 0.32 mm × 0.25 µm film thickness). Injector and detector temperatures were 250 °C and 300 °C, respectively. After injection, the column temperature was programmed to increase to 200 °C for 10 min, subsequently increased to 240 °C at the rate of 3 °C min^−1^, and then held at the final temperature for 4 min. Helium was used as a carrier gas (3 mL min^−1^). The amounts of fatty acids were calculated from the chromatograms and from an internal standard containing FAME.

#### 2.2.4. Color Measurements

Color parameters (L*, a*, b*) were measured using an X-Rite 8200 colorimeter (X-Rite, Inc., Grand Rapids, MI, USA). Samples for color measurements were 5 cm thick and excised at a depth of 20 mm [49]. Each time before its use, the colorimeter was standardized against a white ceramic calibration. Color measurement followed the Commission Internationale de l’Eclairage (CIE) color convention [50] with outputs of L* (lightness/darkness), a* (red/green) and b* (yellow/blue). The color difference (∆E) between control and test samples during storage was calculated according to AMSA [49] using the following formula:$$\text{∆}E=\sqrt{{\left(∆L\right)}^{2}+{\left(∆a\right)}^{2\;}+{\left(∆b\right)}^{2}}$$

In the interpretation of the results, it was assumed that when 0 < ∆E < 1, the observer does not notice the difference; when 1 < ∆E < 2, only an experienced observer may notice the difference; when 2 < ∆E < 3. 5, an unexperienced observer also notices the difference; when 3.5 < ∆E < 5, a clear difference in color is noticed; and when 5 < ∆E, an observer notices two different colors [51]. 

#### 2.2.5. Microbiological Analyses

The microbiological analyses included the number of lactic acid bacteria (LAB), of *Enterobacteriaceae* (EB) bacteria and of *Escherichia coli* (EC). The analyses were made using the TEMPO^®^ LAB automated microbial counting system (Biomerieux, TEMPO^®^ System, Marcy l’Etoile, France). For microbiological determinations, the original TEMPO^®^ tests were used to determine the number of lactic acid bacteria (TEMPO LAB), of *Enterobacteriaceae* (TEMPO EB) and of *Escherichia coli* (TEMPO EC) in the food products. The incubation conditions used for the TEMPO LAB, TEMPO EB and TEMPO EC tests were: incubation time 40–48 h (LAB), 22–27 h (EB, EC); and temperature of incubation: 37 °C (LAB) and 35 °C (EB, EC). The results are expressed as a log CFU g^−1^. 

#### 2.2.6. Biogenic Amines (BAs) Determination

The BAs extraction process was carried out by homogenizing 5 g of each sausage sample with 25 mL of 10% trichloroacetic acid in a homogenizer (1000 rpm, 1 min, IKA T25D, Staufen, Germany). The homogenate was extracted for 1 h in the temperature 4 ± 1 °C. Next, the samples were centrifuged (3000 rpm, 20 min, 4 °C, MPW 350R, Warsaw, Poland). The supernatants were filtered through a Whatman filter No. 1, passed back through a 0.22 μm nylon filter (Alfachem, Lublin, Poland) and were stored at 4 °C until analysis. The analysis of BAs was performed using an AAA 500 amino acid analyze (Ingos, Praha, Czech Republic), equipped with an Ostion LG ANB ion-exchange column (7 × 0.37 cm, 75 °C). Separation was by a stepwise gradient elution using Na+/K+ citric buffers. Solutions of BAs were prepared with a dilution buffer composed of 1.5 mM NaN3, 197 mM NaCl, and 73 mM citric acid in Milli-Q water. The system consisted of a filling chromatographic column and steel pre-column, two chromatographic pumps for transport of elution buffers and derivatization reagent, a cooled carousel for Eppendorf tubes, a dosing valve, a heat reactor, a Vis detector, and a cooled chamber for derivatization reagent. The volume of the injected sample was 100 µL. The reactor temperature was set to 120 °C. Content of the BA (histamine, tyramine, putrescine, cadaverine, spermidine, agmatine and spermine) was determined with reference to the amine standards, which were supplied by Ingos, Czech Republic. The BA concentrations were reported as mg kg^−1^ of product. 

#### 2.2.7. ABTS*+ Radical Scavenging Activity

ABTS*+ were measured according to the method described by Jung et al. [52], Ferysiuk et al. [53] and Erel [54] with some modifications. The extraction process was carried out by homogenizing 5 g of each sample with 20 mL of ethanol using an IKA ULTRA-TURRAX T25 Basic homogenizer at 10,000 g for 1 min and centrifuged at 3000× *g* for 20 min at 4 °C (MPW 350R, Warsaw, Poland). After this process, supernatants were filtered through a Whatman No 1 filter paper. ABTS was dissolved in distilled water to a 7 mM concentration. The ABTS radical cation was produced by dosing the ABTS stock solution with 2.45 mM potassium persulfate (final concentration) in the dark at room temperature for 12–16 h to allow the completion of radical generation. This solution was then diluted with ethanol so that its absorbance was adjusted to 0.70 ± 0.02 at 734 nm. For ABTS measurement, 50 µL of supernatant was added to 1 mL of ABTS+ solution. For the ABTS estimations, absorbance was measured after 15 min using a UV–vis spectrophotometer (Evolution 300 BB, Thermo Electron Corporation, Madison, England) using ethanol as a blank. The ABTS+ capacity was calculated from a standard curve of Trolox equivalent and expressed as mg per g.

#### 2.2.8. DPPH Radical Scavenging Activity

DPPH radical scavenging activity was estimated according to the method of Blois [55], Jung et al. [52] and Ferysiuk et al. [53] with slight modifications. The extraction process was performed in the same way as for the determination of the antioxidant activity in the case of ABTS. The DPPH radical cation was produced by keeping the DPPH stock solutionin the dark at room temperature for 30 min to allow the completion of radical generation. This solution was then diluted with ethanol so that its absorbance was adjusted to 0.90 ± 0.02 at 517 nm. For DPPH measurement, 1 mL of supernatant was added to 1 mL of DPPH solution. A tube containing 1 mL of ethanol and 1 mL of ethanolic DPPH solution (0.2 mM) served as the control. For the DPPH estimations, absorbance was measured after 6 min using a UV–vis spectrophotometer (Evolution 300 BB, Thermo Electron Corporation, Madison, England) using ethanol as a blank. The DPPH capacity was calculated from a standard curve of Trolox equivalent and expressed as mg per g.

### 2.3. Statistical Analysis

The sausage treatments were replicated twice by producing two different batches. Each sample was analyzed in triplicate. The values of the analyzed variables were presented using the mean ± standard deviation. The normality of the distribution of variables in the studied groups was checked using the Shapiro–Wilk test. The differences between the groups were assessed using the ANOVA (together with Tukey’s post-hoc RIR test), and in the case of failure to meet the conditions for its application, the Kruskal–Wallis test. A significance level of *p* < 0.05 was adopted, indicating the existence of statistically significant differences or relationships. The database and statistical analysis were carried out on the basis of the Statistica 9.1 computer software (StatSoft, Poland).

## 3. Results

### 3.1. Results for Tomato Pomace

#### Antioxidant Activity of TP

Table 1 shows the total content of phenols and antioxidant activity of freeze-dried TP prepared for the use in the recipe of meat products. The antioxidant activity was similar to both ABTS and DPPH radicals and ranged from 0.112 to 0.120 mg Trolox eqv. g^−1^, respectively.

### 3.2. Characteristics of Raw Fermented Sausages

#### 3.2.1. Chemical Composition 

The content of fat, protein, water, collagen and salt in the dry fermented sausages is shown in Table 2. The meat products were characterized by high protein content, ranging from 31.85–33.88%. Statistical analysis showed significant differences (*p* ≤ 0.05) between the sausage groups in terms of their fat content. The highest amount of fat was contained in the sample with 0.5% addition of TP (24.80%), and the lowest in the control sample SK (22.23%). The salt concentration of the four groups of fermented sausages was similar due to the same amount added during production and a similar degree of drying of the products during processing. The moisture content ranged from 36.21% to 37.87%. Moreover, statistically significant differences between the samples are shown in Table 2 (*p* ≤ 0.05).

#### 3.2.2. pH and Water Activity

Table 3 shows the pH values and water activity of the experimental sausages with different levels of TP addition. No significant statistical differences were found between the sausage groups. The values of the physicochemical properties were typical of the fermented products. The sausage samples were characterized by a pH value in the range of 4.68–4.71, while the water activity was in the range of 0.885–0.892.

#### 3.2.3. Fatty Acid Profile

The analysis of the main fatty acid fractions showed a statistically significant effect of TP on the content of MUFA, PUFA, n-6 and n-3 fatty acids (Table 4). Samples of sausages containing TP were characterized by a statistically lower content of these groups of fatty acids compared to the control sample. Additionally, a statistically significant difference (*p* ≤ 0.05) was noted in case of the MUFA content between the sample of STP 1% and that of STP 1.5%. The STP 1% was characterized by a significantly higher MUFA content compared to the sausage with higher level of TP addition. The inverse relationship could be observed for PUFA and n-6 fatty acids. Experimental sausages containing TP in various concentrations were characterized by a statistically significantly (*p* ≤ 0.05) higher content of PUFA and n-6 fatty acids compared to the control sausage sample.

#### 3.2.4. Color Parameters

The results of the color parameters (L*a*b*) are given in Table 5. Regarding the parameter L* (lightness), the statistical analysis showed a statistical difference (*p* ≤ 0.05) between the sausage samples. The control sample was characterized by a significantly higher lightness (by about 5 units) than the sample with the highest concentration of TP. There were statistically significant differences between the sausage samples (*p* ≤ 0.05) in the a* color parameter. The addition of TP caused a significant increase in the redness of the sausages in comparison with the control sample (SK). Thus, along with the increase in the level of TP, the value of the a* color parameter of the sausages increased. The value of the b* color parameter increased with the increasing the level of TP. The total color difference (∆E) between the control sample and the samples with TP addition was the highest in the case of the sausage with the highest level of TP. The ∆E values for the sausage with TP show a clear color change compared to the SK.

#### 3.2.5. Results of Microbiological Analysis

The results of microbiological analyses are presented in Table 6. Statistical analysis showed significant differences in the number of *Eneterobacteriacea* between the samples of dry fermented sausages. It was observed that the number of *Eneterobacteriacea* decreased as the level of added TP increased. Similarly, in the case of LAB bacteria, statistically significant differences (*p* ≤ 0.05) were found between the samples. The sample with 0.5% addition of TP was characterized by a statistically higher number of lactic acid bacteria compared to the control sample and the sample with 1% addition of TP. In general, the number of LABs in all groups of dry fermented sausages was high, which proves that the fermentation was properly carried out, and ranged from 8.57 to 8.77 log CFU g^−1^. Each of the sausage samples contained *E. coli* <10 CFU g^−1^.

#### 3.2.6. Content in Biogenic Amines (BAs) 

Table 7 shows the amount of identified biogenic amines in the experimental dry fermented sausages at the end of production. The presence of six amines was indicated, with tyramine, cadaverine and putrescine being the most abundant. Statistical analysis showed statistically significant differences between the trials (*p* ≤ 0.05) for putrescine, cadaverine and agmatine. The samples with the addition of TP were characterized by a significantly lower concentration of putrescine compared to the control sample. Thus, it was observed that with the increase in the share of TP, the amount of putrescine decreased. On the other hand, opposite relationships were observed in the case of cadaverine. The control sample was characterized by the lowest concentration of this amine, while the sample with 1% addition of TP showed the highest concentration of cadaverine. The content of spermidine and spermine was in the range of 2.70–3.30 mg kg^−1^ and 12.30–15.00 mg kg^−1^, respectively. The total content of biogenic amines expressed as mean values in the dry fermented sausages ranged from 186 mg kg^−1^ for the STP 1.5% sample to 204.30 mg kg^−1^ for the STP 0.5% sample, although no statistically significant differences in the total BA content between the samples were noted.

#### 3.2.7. Antioxidant Activity

The results of the antioxidant activity for the ABTS+ and DPPH radicals are presented in Table 8. Significant differences between the samples of dry fermented sausages were demonstrated. It was observed that with increasing concentration of TP, their antioxidant activity also increased. The antioxidant activity for the ABTS+ radical ranged from 0.0690 mg Trolox equivalent g^−1^ for the control sample to 0.1390 mg Trolox equivalent g^−1^ for the sample with 1.5% TP. Similarly, in the case of antioxidant activity against the DPPH radical, the samples with TP were characterized by significantly higher antioxidant activity compared to the control sample. The addition of tomato pomace in the amount of 1.5% significantly increased the antioxidant activity against the DPPH radical compared to the sausage with 0.5% TP addition.

## 4. Discussion

Tomatoes are considered food with high antioxidant properties, mainly due to the presence of several natural antioxidants, which include e.g., lycopene, ascorbic acid, and phenolic compounds [56,57]. Our results confirmed the data obtained by Vinha et al. [40] who studied the effect of peel and seed removal on the nutritional value and antioxidant activity of tomatoes. The results obtained in our study were also similar to the results of Azabou et al. [46] who compared the antioxidant properties of tomato by-products depending on the solvent used. They showed that the concentration of extracts significantly influences their antioxidant properties. The ethanol extract from TP showed the highest antioxidant activity against the DPPH radical. The percentage of inhibition in the tested extracts, depending on their concentration, was 28–84%. Rehal et al. [58] obtained lower results for tomato pomace; however, all studies confirm that TP has strong antioxidant properties. Fat is also a valuable component of tomatoes. Tomato seeds contain 18–22.5% fat in dry matter. The fat fraction is characterized by a high proportion of unsaturated fatty acids (approximately 80%). Linoleic acid (about 58%) is the dominant one among the identified fatty acids. There are also significant amounts of: oleic, linolenic, palmitic, and stearic acids. Particularly noteworthy is the high content of avenasterol (over 12% of the sterol fraction), which has antioxidant activity [59].

In the current research, the effect of the addition of TP on the quality of raw fermented sausages was assessed. Data from the available literature were taken into account when deciding on the level of TP addition. Depending on the type of meat, researchers added different concentrations of TP, ranging from 0.25% to 7% [60,61,62]. Analyzing their results, it could be observed that these products, regardless of TP concentration, were characterized by an increased share of red color. Additionally, thanks to the lycopene contained in tomato, they showed strong antioxidant properties. In meat products with a concentration of up to 2%, a reduction in lipid oxidation was also observed, but at the 3% level of TP, the opposite trend was observed. The authors also investigated the effect of tomato powder on the sensory properties of meat products. They showed that taste and overall product acceptability improved even when the tomato powder level was as low as or lower than 1.5%. However, as the concentration of the additive increased, the acceptability decreased. Based on the above-mentioned studies, in this experiment it was decided to use the TP level in the range of 0.5–1.5%. In our experiment, it was shown that the addition of TP had a significant effect on the chemical composition of the products. The results obtained differed from other authors whose products were characterized by a lower percentage of protein [62,63]. Nevertheless, they claimed that the high protein content of TP could contribute to an increase in this parameter in a meat product. Our results do not confirm this, however, as it can be seen that the protein content of the sausages with 1.5% addition was similar to that obtained in the control sample. This may therefore indicate that only with a higher concentration of TP could the protein content gradually increase, as in the case of the studies by Savadkoohi et al. [62], in which, from 3% concentration upwards, the protein content in beef sausages started to gradually increase. Similar results were obtained by Ghafouri-Oskuei et al. [63]. In their study, the protein content increased with increasing concentration of tomato powder. Similarly in the studies conducted by Eyiler and Oztan [35], a decrease in the water content in dry fermented sausages was noted with an increase in the concentration of added tomato pomace.

The addition of TP had no effect on the physicochemical parameters of dry fermented sausages (pH and water activity). The pH values obtained in our experiment were lower compared to the results of other authors [30,63,64]. In contrast to our results, Eyiler and Oztan [35] noted a decrease in pH with an increase in the concentration of tomato powder in pork sausages. On the other hand, the water activity of the sausages in our experiment was low, in agreement with the studies by Saksomboon et al. [65].

Meat products provide a significant amount of fat. Research shows that the composition of fatty acids can be varied and depends mainly on the animal’s diet, age, weight, sex or race [66]. In the current study, the content of SFA, MUFA and PUFA in dry fermented sausages was at levels of 41.77–41.94%, 43.68–44.35%, 9.21–9.96%, respectively. Karwowska and Dolatowski [67] obtained slightly lower results, in which the dominant fatty acids were MUFA, then SFA, as in our research. However, in contrast to our research, the authors did not show any effect of the addition of freeze-dried cranberry on the PUFA content. In our study, the addition of tomato pomace had no effect on the SFA content; however, it significantly increased the PUFA content. The samples with the addition of TP were characterized by a significantly higher content of PUFA, which had a positive effect on their nutritional value.

Color is one of the most important indicators of the quality of meat and meat products, as it is closely associated with its freshness [68]. Our research showed that the addition of tomato pomace influenced the color parameters of raw fermented sausages. Similarly, Eyiler and Oztan [35] observed that tomato powder acted as a dye in sausages and increased the a* value in the analyzed samples. In the research conducted by these authors, it was shown that the redness of meat products was in the range of 6.28–13.65, while the lightness (L*) gradually decreased with increasing concentration of tomato powder. In our research, a similar trend was noted, with the exception of products with 0.5% tomato pomace, which were characterized by a slightly higher lightness compared to the control products. Moreover, Savadkoohi et al. [62] confirmed that the redness of beef sausages was significantly influenced by the level of tomato pomace. The total color difference (∆E) recorded in the study by these authors ranged from 0.99 to 3.41 [62]. The sausage samples with a higher TP content were characterized by a higher ∆E compared to the control sample. Thus, the addition of TP significantly affects the color parameters in meat products. A very beneficial phenomenon is the increase in the redness of the products, which thus improves the appearance of the final product and may increase the acceptability of the product by potential consumers. 

In recent years, there has been a growing interest in traditional meat products, but these products are a very good growth medium and can be easily contaminated by microorganisms such as *Enterobacteriaceae*. These are considered to be indicator bacteria for the microbiological quality of food and the hygiene status of the production process [69]. Analyzing the results obtained in our study, it can be concluded that the *Enterobacteriaceae* were present in all fermented sausages and ranged from 1.74–3.15 log CFU g^−1^. The effect of the addition of TP on the reduction of the number of these bacteria in raw fermented sausages was observed, which indicates the antimicrobial properties of TP. Similarly, Borrajo et al. [70], who compared the effect of fortification of dry fermented sausage with *Salvia hispanica* and *Nigella sativa* seeds, confirmed that addition of a higher percentage of seeds also resulted in a decrease in *Enterobacteriaceae*. Bazargani-Gilani et al. [71] found that the addition of pomegranate juice inactivated *Enterobacteriacea*. Lactic acid bacteria are very important in the production of fermented meat products as they are used as starter cultures [72]. During fermentation, the number of lactic acid bacteria increases [73]. Their growth is very beneficial as they can produce a variety of bacteriocins that are generally active against Gram positive foodborne pathogens such as *Listeria monocytogenes*, *Staphylococcus aureus*, *Clostridium perfringens* and *Bacillus cereus* [74]. In the sausages produced in this study, the presence of LAB ranged from 8.57–8.77 log CFU g^−1^. There was no significant effect of the addition of TP on the LAB number in fermented sausages. Borrajo et al. [70] indicated that the addition of chia seeds or black cumin increased the proliferation of LAB. The antimicrobial activity of natural extracts results mainly from the content of phenolic compounds [56].

Both fermented and protein-rich foods are very susceptible to the formation of biogenic amines. These are produced by bacterial decarboxylation of appropriate amino acids in food, and their concentration in fermented food products depends on several factors, including the hygiene of raw materials, microbiological status, fermentation status and duration of fermentation [75]. Some authors have stated that tyramine, putrescine and cadaverine are the most dominant types of amines found in fermented foods, including raw fermented sausages [75,76]. The present studies confirmed this relationship as tyramine, putrescine and cadaverine were dominant in the experimental raw fermented sausages. Particular attention should be paid to tyramine, as it is one of the main biogenic amines associated with certain health disorders, including vasoactive and psychoactive reactions [77,78]. In fermented sausages tyramine is produced mainly by lactic acid bacteria [78]. For this reason, sausage samples were characterized by a high level of this compound, as the number of lactic acid bacteria increases as a result of fermentation [79]. Comparing the obtained results with the results of other authors, it can be concluded that they noticed similar observations. De Mey et al. [80], who compared the content of biogenic amines in commercial dry fermented sausages, found that tyramine is the main biogenic amine. The concentration of this compound ranged from 3.6 to 149.9 mg kg^−1^. Similarly to our findings, they also observed low concentrations of spermidine and spermine in the tested meat products. Similar results were also obtained by Borrajo et al. [70]. However, these authors noticed the opposite trend to that observed in our research in the content of spermine, which was present in the smallest amount in the samples of sausages with the addition of chia seeds and black cumin. The results of this study did not show any significant effect of tomato pomace on spermine content. In the context of the exposure of fermented meat products to the presence of biogenic amines, the challenge for the meat industry is the development of technology for obtaining products free or almost free of these compounds [81]. 

The development of the functional food market is closely related to the use of bioactive ingredients useful for the development of innovative products. In this context, much attention has recently been given to natural compounds derived from plant waste and their relationship to high bioactivity [82]. In the present study, it was observed that with the increase in TP concentration in fermented sausages, their antioxidant activity also increases. These observations are consistent with the results of other authors. Riazi et al. [83], who investigated the effect of grape pomace on meat products with a reduced content of nitrites, showed that the samples with the addition of pomace showed higher antioxidant activity compared to the control samples. As in our findings, the antioxidant activity increased with increasing grape pomace content. A similar tendency was also noticed by Ramli et al. [84] who studied the effect of powdered passion fruit extract on the antioxidant effect of preserved meat products. Based on the results obtained, it can be concluded that tomato pomace can be a potential component of functional food. Thanks to its antioxidant properties, it can be used as a natural preservative for meat products. Additional post-storage studies should be performed to confirm the effect of the storage period on the values of the antioxidant activities of the fermented sausages.

## 5. Conclusions

In summary, the results found in this article showed that the addition of TP to raw fermented sausages effectively increased the antioxidant potential of the product, and it can therefore be assumed that it enriched the products with phenolic compounds. The antioxidant activity of sausages was closely related to the concentration of tomato pomace with which they were enriched. The product containing 1.5% TP was characterized by the strongest antioxidant properties. Increasing the addition of TP also resulted in increased redness of the meat product, which can be assumed to have a positive effect on consumer acceptability. Moreover, the samples of sausages with the addition of TP were characterized by a lower number of *Enterobacteriaceae*. Taking into account the results obtained, tomato pomace can be used as a natural additive in the production of raw fermented sausages and thus enable a reduction in nitrite content. Being an alternative to nitrogen compounds, the opportunity thus created to use this by-product of tomato processing has the added advantage of reducing food losses. The inclusion of TP—rich in bioactive compounds—in meat products is also an excellent strategy for the development of innovative meat products with increased nutritional value and an improved nutritional profile. The most promising results were obtained for the meat products with 1.5% addition of TP. Future studies are necessary to investigate the effect of a higher level of TP addition on the quality of raw fermented sausage, with particular regard to the sensory evaluation.

## Figures and Tables

**Figure 1 biomolecules-12-01695-f001:**

Cross-sectional appearance of fermented sausage: (**a**) control sample; (**b**) sample with 0.5% addition of TP; (**c**) sample with 1% addition of TP; and (**d**) sample with 1.5% addition of TP.

**Table 1 biomolecules-12-01695-t001:** Antioxidant activity and TPC of freeze-dried TP.

Properties	Freeze-Dried TP
DPPH [mg Trolox eqv. g^−1^]	0.120 ± 0.004
ABTS [mg Trolox eqv. g^−1^]	0.112 ± 0.007
TPC [mg gallic acid eqv. g^−1^]	4.080 ± 0.167

**Table 2 biomolecules-12-01695-t002:** Proximate chemical composition (%) of dry fermented sausages.

Compound	SK	STP 0.5%	STP 1%	STP 1.5%
Fat	22.23 ± 0.02 a	24.80 ± 0.01 c	24.46 ± 0.01 d	23.00 ± 0.02 b
Protein	33.88 ± 0.02 c	31.85 ± 0.02 a	31.88 ± 0.02 a	33.74 ± 0.01 b
Moisture	37.87 ± 0.10 c	37.10 ± 0.04 b	37.04 ± 0.07 b	36.21 ± 0.11 a
Collagen	2.54 ± 0.23 a	2.74 ± 0.46 ab	3.21 ± 0.22 ab	3.39 ± 0.29 b
Salt	4.00 ± 0.15 a	3.90 ± 0.04 a	3.82 ± 0.12 a	3.80 ± 0.09 a

SK—control sample; STP 0.5%—sample with 0.5% addition of TP; STP 1%—sample with 1% addition of TP; STP 1.5%—sample with 1.5% addition of TP. Means with different lowercase letters (a–d) differ significantly (*p* ≤ 0.05).

**Table 3 biomolecules-12-01695-t003:** pH and water activity of dry fermented sausages.

Properties	SK	STP 0.5%	STP 1%	STP 1.5%
pH	4.71 ± 0.14	4.68 ± 0.01	4.71 ± 0.01	4.71 ± 0.02
Water activity	0.888 ± 0.006	0.892 ± 0.002	0.890 ± 0.003	0.885 ± 0.007

No significant differences between samples were found. SK—control sample; STP 0.5%—sample with sample with 0.5% addition of TP; STP 1%—sample with 1% addition of TP; STP 1.5%—sample with 1.5% addition of TP.

**Table 4 biomolecules-12-01695-t004:** Main fractions of the fatty acid profile (g/100 g) of dry fermented sausages.

Compound	SK	STP 0.5%	STP 1%	STP 1.5%
SFA	9.43 ± 0.08 b	10.44 ± 0.08 c	10.3 ± 0.11 c	8.1 ± 0.13 a
MUFA	9.98 ± 0.11 b	10.97 ± 0.07 c	10.83 ± 0.16 c	8.44 ± 0.11 a
PUFA	2.07 ± 0.05 b	2.48 ± 0.02 c	2.45 ± 0.09 c	1.91 ± 0.01 a
n-3	0.09 ± 0.01 a	0.12 ± 0.01 b	0.12 ± 0.01 b	0.09 ± 0.00 a
n-6	1.98 ± 0.04 b	2.35 ± 0.01 c	2.33 ± 0.09 c	1.82 ± 0.01 a

SK—control sample; STP 0.5%—sample with 0.5% addition of TP; STP 1%—sample with 1% addition of TP; STP 1.5%—sample with 1.5% addition of TP. Means with different lowercase letters (a–c) differ significantly (*p* ≤ 0.05).

**Table 5 biomolecules-12-01695-t005:** CIE L*, a* and b* color parameters of dry fermented sausages.

Color Parameter	SK	STP 0.5%	STP 1%	STP 1.5%
L*	50.12 ± 5.09 ab	51.63 ± 3.36 b	49.77 ± 2.78 ab	45.55 ± 1.21 a
a*	9.76 ± 2.15 a	11.48 ± 2.38 ab	13.31 ± 1.45 bc	15.72 ± 0.84 c
b*	6.64 ± 1.18 a	9.64 ± 1.81 b	10.73 ± 1.36 bc	12.36 ± 1.04 c
∆E		3.77	5.41	9.49

SK—control sample; STP 0.5%—sample with 0.5% addition of TP; STP 1%—sample with 1% addition of TP; STP 1.5%—sample with 1.5% addition of TP. Means with different lowercase letters (a–c) differ significantly (*p* ≤ 0.05).

**Table 6 biomolecules-12-01695-t006:** The results of microbiological analyses of dry fermented sausages.

Bacteria	SK	STP 0.5%	STP 1%	STP 1.5%
*Enterobacteriaceae* [log CFU g^−1^]	3.02 ± 0.06 c	3.15 ± 0.09 c	2.46 ± 0.15 b	1.74 ± 0.22 a
*Lactic acid bacteria* [log CFU g^−1^]	8.60 ± 0.06 ab	8.77 ± 0.01 c	8.57 ± 0.05 a	8.74 ±0.08 bc
*E. coli* [log CFU g^−1^]	<10	<10	<10	<10

SK—control sample; STP 0.5%—sample with 0.5% addition of TP; STP 1%—sample with 1% addition of TP; STP 1.5%—sample with 1.5% addition of TP. Means with different lowercase letters (a–c) differ significantly (*p* ≤ 0.05).

**Table 7 biomolecules-12-01695-t007:** The biogenic amines of dry fermented sausages [mg kg^−1^].

Compound	SK	STP 0.5%	STP 1%	STP 1.5%
Tyramine	38.00 ± 3.00 a	41.30 ± 1.50 a	45.70 ± 1.20 a	38.30 ± 4.70 a
Putrescine	53.30 ± 5.80 c	37.00 ± 3.00 b	26.3 ± 1.20 a	24.3 ± 1.50 a
Cadaverine	86.70 ± 4.60 a	105.00 ± 2.60 b	110.70 ± 6.80 b	104.30 ± 3.10 b
Spermidine	2.70 ± 0.60 a	3.00 ± 0.00 a	3.00 ± 0.00 a	3.30 ± 0.60 a
Agmatine	0.70 ± 1.20 a	4.30 ± 0.60 ab	5.30 ± 0.60 b	3.30 ± 2.90 ab
Spermine	15.00 ± 1.00 a	13.70 ± 1.50 a	12.30 ± 3.80 a	13.00 ± 2.60 a
Total	197.00 ± 13.20 a	204.30 ± 9.50 a	203.30 ± 8.00 a	186.00 ± 7.50 a

SK—control sample; STP 0.5%—sample with 0.5% addition of TP; STP 1%—sample with 1% addition of TP; STP 1.5%—sample with 1.5% addition of TP. Means with different lowercase letters (a–c) differ significantly (*p* ≤ 0.05).

**Table 8 biomolecules-12-01695-t008:** Antioxidant activity of dry fermented sausages.

Properties	SK	STP 0.5%	STP 1%	STP 1.5%
DPPH [mg Trolox eqv. g^−1^]	0.069 ± 0.006 a	0.085 ± 0.004 b	0.095 ± 0.003 bc	0.102 ± 0.001 c
ABTS [mg Trolox eqv. g^−1^]	0.069 ± 0.001 a	0.102 ± 0.001 b	0.121 ± 0.001 c	0.139 ± 0.001 d

SK—control sample; STP 0.5%—sample with 0.5% addition of TP; STP 1%—sample with 1% addition of TP; STP 1.5%—sample with 1.5% addition of TP. Means with different lowercase letters (a–d) differ significantly (*p* ≤ 0.05).

## Data Availability

Not applicable.

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
