# Peer review of "Fatty Acids Profile and Antioxidant Properties of Raw Fermented Sausages with the Addition of Tomato Pomace"

_biomolecules, 2022, doi:10.3390/biom12111695_

Round 1
Reviewer 1 Report
Using four different sausage formulations, the objective of the present study was to examine the effect of tomato pomace on the physicochemical parameters and fatty acid profile as well as on the antioxidant activity of dry-fermented sausages produced with reduced nitrite content. The manuscript is overall well written and therefore deserves to be published in the international journal Biomolecules once the few elements mentioned below have been considered:
· Page 1 / Line 29: This first sentence is too general and needs to be clarified. On what scale is the level of meat consumption constantly increasing? In the world? In the authors' country, i.e. Poland? Elsewhere?
· Page 4/Lines 184-186: Please justify the authors' choice of tomato pomace content: why 0.5%; 1% and 1.5%. Why not more?
· Page 6/Line 289 & Page 7/Line 307: Please write "the capacity ... was calculated from ...".
· Page 7/Line 323: This is table 1, not 2.
· Page 10/Section 3.2.7: Antioxidant activity results should be presented in Table 8, not 9. Please correct both in the text and in the table legend.
· Page 12/Line 529: It is not clear whether "recorded in this study" refers to the authors' own work presented in the manuscript or to the results of the work related to reference 65 cited in the previous sentence. Please rewrite the sentence to remove this ambiguity.
· Discussion section: I think it would have been relevant to discuss the tomato pomace contents used by the authors of the article in relation to the contents that might have been used by other authors in different works.
· It is always awkward in a scientific article to state and conclude that the best or most interesting result was obtained for one of the extreme values of the field of study, in this case: 1.5% tomato pomace. This raises the question whether an even better result could have been obtained if the addition of tomato pomace had been higher: 1.8%, 2.0% or even more. This point should be mentioned or even discussed somewhere in the manuscript.
· Finally, no information is given on the organoleptic qualities of the dry fermented sausages produced following the addition of tomato pomace. If this aspect was not the focus of the work presented here, it deserves to be mentioned at least in the perspectives part of this work.
Author Response
REVIEWER 1: Review Report (Round 1)
The authors thank for the valuable comments of the Reviewer, which allowed for the improvement of the article. Changes to the notes have been made to the text in BLUE.
General comments:
Using four different sausage formulations, the objective of the present study was to examine the effect of tomato pomace on the physicochemical parameters and fatty acid profile as well as on the antioxidant activity of dry-fermented sausages produced with reduced nitrite content. The manuscript is overall well written and therefore deserves to be published in the international journal Biomolecules once the few elements mentioned below have been considered:
Specific comments
- Page 1, line 29: This first sentence is too general and needs to be clarified. On what scale is the level of meat consumption constantly increasing? In the world? In the authors' country, i.e. Poland? Elsewhere?
According to Reviewer’s recommendation the first sentence has been reformulated and completed (lines 29-30)
- Page 4, lines 184-186: Please justify the authors' choice of tomato pomace content: why 0.5%; 1% and 1.5%. Why not more?
This part of manuscript was supplemented with the sentences „Data from the available literature were taken into account when deciding on the level of TP addition. Depending on the type of meat, researchers added different concentrations of TP, ranging from 0.25% to 7% [64,65,66]. Analyzing their results, it could be observed that these products, regardless of TP concentration, were characterized by an increased share of red color. Additionally, thanks to the lycopene contained in tomato, they showed strong antioxidant properties. In meat products with a concentration of up to 2%, a reduction in lipid oxidation was also observed, but at the 3% level of TP, the opposite trend was observed. The authors also investigated the effect of tomato powder on the sensory properties of meat products. They showed that taste and overall product acceptability changed even when the tomato powder level was 1.5%. However, as the concentration of the additive increased, the acceptability decreased. Based on the above-mentioned studies, in this experiment it was decided to use the TP level in the range of 0.5 - 1.5%” (lines 483-495).
- Page 6, line 289 & Page 7/Line 307: Please write "the capacity ... was calculated from ...".
According to Reviewer’s recommendation the sentence “The ABTS+ capacity was calculated from a standard curve of Trolox equivalent and expressed as mg per g“ and “DPPH capacity was calculated from a standard curve of Trolox equivalent and expressed as mg per g” was written (lines 280-282 and 294-295).
- Page 7, line 323: This is table 1, not 2.
I am sorry for the mistake, the table numbers have been verified.
- Page 10/ Section 3.2.7: Antioxidant activity results should be presented in Table 8, not 9. Please correct both in the text and in the table legend.
Sorry for the mistake the table numbers have been verified.
- Page 12/Line 529: It is not clear whether "recorded in this study" refers to the authors' own work presented in the manuscript or to the results of the work related to reference 65 cited in the previous sentence. Please rewrite the sentence to remove this ambiguity.
The sentence has been changed to “The total color difference (∆E) recorded in the study by these authors ranged from 0.99 to 3.41 [67]” (lines 536-537).
- Discussion section: I think it would have been relevant to discuss the tomato pomace contents used by the authors of the article in relation to the contents that might have been used by other authors in different works.
This part of manuscript has been supplemented according to Reviewer’s recommendation (lines 483-495)
- It is always awkward in a scientific article to state and conclude that the best or most interesting result was obtained for one of the extreme values of the field of study, in this case: 1.5% tomato pomace. This raises the question whether an even better result could have been obtained if the addition of tomato pomace had been higher: 1.8%, 2.0% or even more. This point should be mentioned or even discussed somewhere in the manuscript.
The sentences „The most promising results were obtained for the meat products with 1.5% addition of TP. Future studies are necessary to investigate the effect of a higher level of TP addition on the quality of raw fermented sausage, with particular regard to the sensory evaluation” were added to the Conclusion (lines 622-625).
- Finally, no information is given on the organoleptic qualities of the dry fermented sausages produced following the addition of tomato pomace. If this aspect was not the focus of the work presented here, it deserves to be mentioned at least in the perspectives part of this work.
The sentence „Future studies are necessary to investigate the effect of a higher level of TP addition on the quality of raw fermented sausage, with particular regard to the sensory evaluation” was added to the Conclusion (lines 622-625).
Reviewer 2 Report
I am not convinced that the choice of 'Biomolecules' journal is right, rather 'Foods' should be chosen. The whole thing is based on the influence of tomato pomace on the physicochemical parameters, fatty acid, biogenic amines profile, and antioxidant properties of dry fermented sausages with reduced nitrite content. Additionally, microbiological analyzes were performed. The presented biomolecules do not explain their function, they are typical parameters that have been analyzed in raw fermented sausages with the addition of tomato pomace. For example, fatty acids are presented in groups (SFA, MUFA, PUFA). There is no in-depth statistical analysis, such as PCA. The Discussion did not compare the obtained results with world standards, e.g., European standards for food.
Summarizing, the manuscript is well-organized and well-written; however, the Abstract is not clearly presented. The abbreviation: ABTS and DPPH should be explained.
I leave the final decision to the editors.
Author Response
REVIEWER 2: Review Report (Round 1)
The authors thank for the valuable comments of the Reviewer, which allowed for the improvement of the article. Changes to the notes have been made to the text in green.
I am not convinced that the choice of 'Biomolecules' journal is right, rather 'Foods' should be chosen. The whole thing is based on the influence of tomato pomace on the physicochemical parameters, fatty acid, biogenic amines profile, and antioxidant properties of dry fermented sausages with reduced nitrite content. Additionally, microbiological analyzes were performed. The presented biomolecules do not explain their function, they are typical parameters that have been analyzed in raw fermented sausages with the addition of tomato pomace. For example, fatty acids are presented in groups (SFA, MUFA, PUFA). There is no in-depth statistical analysis, such as PCA. The Discussion did not compare the obtained results with world standards, e.g., European standards for food.
Summarizing, the manuscript is well-organized and well-written; however, the Abstract is not clearly presented. The abbreviation: ABTS and DPPH should be explained.
I leave the final decision to the editors.
Thank you for valuable comments. Based on them, we reformulated Abstract (lines 19-25), explained ABTS and DPPH (lines 103-104).
Reviewer 3 Report
Dear Editor,
Thank you very much for giving me the possibility to read the paper entitled “Fatty acids profile and antioxidant properties of raw fermented sausages with the addition of tomato pomace”. It is interesting and well conducted, however some major concerns are reported below.
Introduction
The introduction is a little bit wordy, and does not focus well on the background of the topic.
M&M
L 111 – Which variety? Better to specify
L 114 – Time? There is a standard time or at minimum time?
L118 – why the acronym is reported in the subtitle and not in the text after?
L119-123 – not clear. Analysis conducted on tomato pomace or on dried tomato pomace?
L129 – here and elsewhere (L135, L141 and others) check the space
L192 – 17 days for all or at minimum?
L310-318 – Not clear. How many samples? I understand 4 thesis, 2 samples for each one and analysis in triplicate. How did you calculate the normality distribution? Please explain better the n of samples for each experimental trial
Table 2 - How do you explain differences in fat content?
Table 4 – Considering differences in fat content probably better to report FA in g/100g of product, not in % on total FAME
Table 5 – About lightness, please check superscripts for statistical analysis, probably not correct.
Author Response
REVIEWER 3: Review Report (Round 1)
The authors thank for the valuable comments of the Reviewer, which allowed for the improvement of the article. Changes to the notes have been made to the text in orange.
Dear Editor,
Thank you very much for giving me the possibility to read the paper entitled “Fatty acids profile and antioxidant properties of raw fermented sausages with the addition of tomato pomace”. It is interesting and well conducted, however some major concerns are reported below.
Introduction
The introduction is a little bit wordy, and does not focus well on the background of the topic.
According to Reviewer’s recommendation, the Introduction has been shortened a bit.
M&M
L 111 – Which variety? Better to specify
The sentence was changed to „The material for the research was one variety of tomatoes (Solanum lycopersicum L.) purchased at a local supermarket” (line 108).
L 114 – Time? There is a standard time or at minimum time?
Fresh tomato pomace was placed in a freeze dryer and freeze-dried until a constant weight was obtained, the treatment took about 4 days.
L118 – why the acronym is reported in the subtitle and not in the text after?
Corrections to this section have been implemented.
L119-123 – not clear. Analysis conducted on tomato pomace or on dried tomato pomace?
The sentence „Thus prepared, freeze-dried TP (called TP) was analyzed in accordance with the following methods” was added to 2.1. section (lines114-115).
L129 – here and elsewhere (L135, L141 and others) check the space
The corrections were introduced in the text.
L192 – 17 days for all or at minimum?
The sausages were kept in the fermentation chamber until the weight loss was 30%±3%. After 17 days, all sausages reached the assumed loss.
L310-318 – Not clear. How many samples? I understand 4 thesis, 2 samples for each one and analysis in triplicate. How did you calculate the normality distribution? Please explain better the n of samples for each experimental trial.
The sausage treatments were replicated twice by producing two different batches. Each sample was analyzed in triplicate. The normality of the distribution of variables in the studied groups was checked using the Shapiro-Wilk test. The section 2.3 has been clarified (lines 297-298).
Table 2 - How do you explain differences in fat content?
It is difficult for us to explain the differences because the raw material composition was the same for all samples. The results for the replicates were very similar, therefore the differences were statistically significant.
Table 4 – Considering differences in fat content probably better to report FA in g/100g of product, not in % on total FAME
Table 4 has been corrected and the fatty acid profile is presented as suggested in g / 100g.
Table 5 – About lightness, please check superscripts for statistical analysis, probably not correct.
We made an adjustment and verified the data in Table 5.
Reviewer 4 Report
The manuscript entitled “Fatty acids profile and antioxidant properties of raw fermented sausages with the addition of tomato pomace” is finely planned (but with some defaults), methods are adequately described, results clearly presented, and the discussion supported by results.
Under my point of view, there is a great flaw in this work. The authors studied the composition of fatty acids (FA) of sausages but tomato pomace (TP) was not analyzed. Tomato seeds contain about 25% fat and it is especially rich in linoleic and oleic acids. Without having studied tomato FA, how can they explain the changes in FA observed in sausages? The authors have studied only the phytochemical composition of TP (carotene, lycopene and total phenols), so they can discuss the same parameters in sausages and how TP influences color and fermentation, but never FA composition. Besides, carotene, lycopene and total phenols were not analyzed in sausages. The analyses from both materials (TP and sausages) are not consistent. In my opinion, there are two options. The first one is to complete the analysis and present the FA composition of TP and the phytochemicals (carotene, lycopene and total phenols) from sausages. If this is not possible, just delete the results and discussion of the changes in FA composition of sausages because the authors did not have any information to explain these changes, and they must also delete the content in carotene, lycopene and total phenols of TP. If these parts are deleted, the total number of references will decrease, because I think that there are too many references for a research work. In the present form the manuscript is not acceptable for publication.
In addition, there are other minor changes to be done:
· - Change “tomato pomace” by “TP” all over the text
· - Lines 129-131: there are different wavelength for the DPPH assay
· - Line 130: 300 microliters
· - Line 231: reference cited without number (CIE, 1978)
· - The extraction process for the determination of DPPH and ABTS assays from sausages is the same for both assays (lines 274-278 for ABTS and lines 293-297 for DPPH). Do not repeat.
· - Line 519: References numbered as 78 and 79 are wrong. I think they must be 73 and 74.
· - Line 611: in the conclusions, the authors said that “the samples of sausages with the addition of TP were characterized by a lower content of biogenic amines” but this is not true. Only 1.5% sausages had lower total amounts than SK. Another thing to discuss is if the differences presented are significant or not. In my opinion, the results in Table 7 should be presented as total amount in mg/kg with the level of significance, and the different amines as % on the total. Looking at the results, probably there are not significant differences among samples but only differences in composition.
· - Some references are not coincident in text and in list (Azabou/Azaboua, Vinh/Vinha), and other ones are repeated (7 and 11 are the same, and 31-36). The use of software such as Zotero, EndNote, Reference Manager, … to manage the bibliographic references is clearly recommended.
Author Response
REVIEWER 4: Review Report (Round 1)
The authors thank for the valuable comments of the Reviewer, which allowed for the improvement of the article. Changes to the notes have been made to the text in red.
The manuscript entitled “Fatty acids profile and antioxidant properties of raw fermented sausages with the addition of tomato pomace” is finely planned (but with some defaults), methods are adequately described, results clearly presented, and the discussion supported by results.
Under my point of view, there is a great flaw in this work. The authors studied the composition of fatty acids (FA) of sausages but tomato pomace (TP) was not analyzed. Tomato seeds contain about 25% fat and it is especially rich in linoleic and oleic acids. Without having studied tomato FA, how can they explain the changes in FA observed in sausages? The authors have studied only the phytochemical composition of TP (carotene, lycopene and total phenols), so they can discuss the same parameters in sausages and how TP influences color and fermentation, but never FA composition. Besides, carotene, lycopene and total phenols were not analyzed in sausages. The analyses from both materials (TP and sausages) are not consistent. In my opinion, there are two options. The first one is to complete the analysis and present the FA composition of TP and the phytochemicals (carotene, lycopene and total phenols) from sausages. If this is not possible, just delete the results and discussion of the changes in FA composition of sausages because the authors did not have any information to explain these changes, and they must also delete the content in carotene, lycopene and total phenols of TP. If these parts are deleted, the total number of references will decrease, because I think that there are too many references for a research work. In the present form the manuscript is not acceptable for publication.
The aim of the study was to assess the effect of the addition of Tomato pomace (rich in lycopene, polyphenols, carotene) on the quality of fermented sausages, which is the result of changes taking place during production (fermentation and maturation). Changes during the production of sausages are conditioned by the characteristics of TP. Therefore, TP was characterized first, and then the characteristics of the sausages, which could have changed as a result of the influence of TP. Our main goal was to demonstrate the antioxidant and antimicrobial properties of the used Tomato pomace. We really care about leaving the results posted, we ask the Reviewer for permission to do so. At the same time, the discussion has been completed in the context of the presence of fat in TP based on the literature data (lines 447-452).
In addition, there are other minor changes to be done:
Change “tomato pomace” by “TP” all over the text
As suggested, we have changed "tomato pomace" throughout the text to the abbreviation "TP".
Lines 129-131: there are different wavelength for the DPPH assay
The changes has been made to this part of manuscript (line 129).
Line 130: 300 microliters
The changes has been made to this part of manuscript (127-128).
Line 231: reference cited without number (CIE, 1978)
The changes has been made to this part of manuscript (line 226).
The extraction process for the determination of DPPH and ABTS assays from sausages is the same for both assays (lines 274-278 for ABTS and lines 293-297 for DPPH). Do not repeat.
The changes has been made to this part of manuscript according to Reviewer’s recommendation (lines 285-287).
Line 519: References numbered as 78 and 79 are wrong. I think they must be 73 and 74.
The changes has been made according to Reviewer’s recommendation. The References has been verified.
Line 611: in the conclusions, the authors said that “the samples of sausages with the addition of TP were characterized by a lower content of biogenic amines” but this is not true. Only 1.5% sausages had lower total amounts than SK. Another thing to discuss is if the differences presented are significant or not. In my opinion, the results in Table 7 should be presented as total amount in mg/kg with the level of significance, and the different amines as % on the total. Looking at the results, probably there are not significant differences among samples but only differences in composition.
The authors apologize for any mistakes in the article. They have been verified. In line with the valuable comments of the Reviewer, an additional column was added in Table 7, indicating the total content of amines expressed in mg / kg. The description in this regard has also been supplemented in the Results section (lines 408-411).
Some references are not coincident in text and in list (Azabou/Azaboua, Vinh/Vinha), and other ones are repeated (7 and 11 are the same, and 31-36). The use of software such as Zotero, EndNote, Reference Manager, … to manage the bibliographic references is clearly recommended.
We have improved both the references to the literature and the bibliography throughout the manuscript.
Round 2
Reviewer 3 Report
Dear Editor,
authors' efforts made this paper ready for pubblication.
Author Response
Response: The authors thank for the valuable comments of the Reviewer, which allowed for the improvement of the article.
Reviewer 4 Report
I would thank the authors for their effort in manuscript corrections. However, I think that it is not acceptable for printing in its present form.
· The first thing is the TP carotenoid determination and discussion. These results are not relevant for the main objective of the work and, therefore, they must be deleted (epigraph 2.1.2 from material and methods, table 1 partially, lines 453-481 in Discussion, and bibliographic references)
· Reference 68 cited in lines 507 and 512 is, in fact, number 69
· In Conclusions, it is said “sausages with the addition of TP were characterized by a lower content of biogenic amines” but this is not true. The total content did not change significantly, only the percentages of different amines.
Author Response
REVIEWER 4: Review Report (Round 2)
I would thank the authors for their effort in manuscript corrections. However, I think that it is not acceptable for printing in its present form.
Response: The authors thank for the valuable comments of the Reviewer, which allowed for the improvement of the article.
The first thing is the TP carotenoid determination and discussion. These results are not relevant for the main objective of the work and, therefore, they must be deleted (epigraph 2.1.2 from material and methods, table 1 partially, lines 453-481 in Discussion, and bibliographic references)
Response: According to Reviewer’s recommendation, methods, results and discussion related to carotenoids have been delated. Bibliographic references have been verified.
Reference 68 cited in lines 507 and 512 is, in fact, number 69’
Response: I am very sorry for the mistake. It has been corrected.
In Conclusions, it is said “sausages with the addition of TP were characterized by a lower content of biogenic amines” but this is not true. The total content did not change significantly, only the percentages of different amines.
Response: I am very sorry for the mistake. This sentence has been delated.